# Temporal and Locational Values of Images Affecting the Deep Learning of Cancer Stem Cell Morphology

**DOI:** 10.3390/biomedicines10050941

**Published:** 2022-04-19

**Authors:** Yumi Hanai, Hiroaki Ishihata, Zaijun Zhang, Ryuto Maruyama, Tomonari Kasai, Hiroyuki Kameda, Tomoyasu Sugiyama

**Affiliations:** 1School of Bioscience and Biotechnology, Tokyo University of Technology, 1401-1 Katakura-machi, Hachioji, Tokyo 192-0982, Japan; b01172095b@edu.teu.ac.jp (Y.H.); b011830622@edu.teu.ac.jp (Z.Z.); maruyamart@stf.teu.ac.jp (R.M.); 2School of Computer Science, Tokyo University of Technology, 1401-1 Katakura-machi, Hachioji, Tokyo 192-0982, Japan; ishihata@stf.teu.ac.jp (H.I.); kameda@stf.teu.ac.jp (H.K.); 3Neutron Therapy Research Center, Okayama University, 2-5-1 Shikada-cho, Kita-ku, Okayama 700-8558, Japan; t-kasai@okayama-u.ac.jp

**Keywords:** artificial intelligence, segmentation, classification, cancer stem cell, cell morphology

## Abstract

Deep learning is being increasingly applied for obtaining digital microscopy image data of cells. Well-defined annotated cell images have contributed to the development of the technology. Cell morphology is an inherent characteristic of each cell type. Moreover, the morphology of a cell changes during its lifetime because of cellular activity. Artificial intelligence (AI) capable of recognizing a mouse-induced pluripotent stem (miPS) cell cultured in a medium containing Lewis lung cancer (LLC) cell culture-conditioned medium (cm), miPS-LLCcm cell, which is a cancer stem cell (CSC) derived from miPS cell, would be suitable for basic and applied science. This study aims to clarify the limitation of AI models constructed using different datasets and the versatility improvement of AI models. The trained AI was used to segment CSC in phase-contrast images using conditional generative adversarial networks (CGAN). The dataset included blank cell images that were used for training the AI but they did not affect the quality of predicting CSC in phase contrast images compared with the dataset without the blank cell images. AI models trained using images of 1-day culture could predict CSC in images of 2-day culture; however, the quality of the CSC prediction was reduced. Convolutional neural network (CNN) classification indicated that miPS-LLCcm cell image classification was done based on cultivation day. By using a dataset that included images of each cell culture day, the prediction of CSC remains to be improved. This is useful because cells do not change the characteristics of stem cells owing to stem cell marker expression, even if the cell morphology changes during culture.

## 1. Introduction

A minor population of tumor cells promotes tumor growth. The cell labeled as cancer stem cell (CSC) depicts the analogy between the renewal of normal and tumor tissues through each activity of the tissue specific to the adult stem cell [1]. Recent CSC research has provided new insights on CSC properties. The generation of CSC happens not only by self-renewal but also by phonotype transition from non-CSC. CSC and non-CSC adopting a quiescent state often acquire chemoresistance and radiation resistance. Therefore, the development of new therapies targeting these cells is required.

Cell morphology is associated with stem cell (SC) property. Non-SC fibroblasts change their morphology during the induction of stem cells by Yamanaka’s factors [2]. Mouse iPS-derived CSCs expressing an SC marker formed spherical colonies while non-CSCs without the expression failed to do so [3]. Mesenchymal SCs lose morphological features by cocultivation with lung cancer cells, while increasing the colony-forming capability [4]. It is commendable for SC biologists to notice these morphological alterations using microscopy. However, it is not easy to be aware of every moment for observing the cells. Some SCs or CSCs might lose the SC character for unknown reasons during culture. Considering the importance of morphological characteristics of CSC, we studied the possibility of an AI to recognize the CSC morphology [5].

Recently, AI technology was applied in cell biology to learn novel skills of a trainee with expertise, such as, for the evaluation of human-induced pluripotent stem cell-derived cardiomyocyte (hiPSC-CM) culture into bad and good conditions, as performed through inspection by a well-trained experimenter. Accordingly, a trained AI was used to classify the two groups using CNN [6]. The induction of mouse embryonic SC differentiation in a culture was performed to make groups of pluripotent and differentiated cells. Subsequently, a trained AI was used to recognize undifferentiated cells from differentiated cells with high accuracy [7]. Categorizing single cells in leukemia smear images of patients and healthy participants immunologically; then, a trained AI was used to classify these into six classes of cell types using several CNN frameworks [8]. A trained AI using automatically classified image sets showed more excellent segmentation capability of cultured hiPSC image into five morphologically different cell types than manual segmentation by experts [9]. A trained neural SC bright-field image in suspension was used to enable the AI to predict among neurons, astrocytes, and oligodendrocytes [10]. Image-based phenotyping with deep learning could determine the morphological cellular phenotype that has been previously unexplored [11]. However, these AI models still require improvements in terms of labeling or identifying the cells of the AI output among various cell types in an image because specification of the cells in an image is important in cell biology.

The AI can segment cells with the same morphologic features from other cell types in cases where clear distinctions among the various types exist, such as flat and square shapes. So far, we studied the possibility of application of the AI to segment cells with the unrecognized feature of CSC morphology using CGAN [5]. Although it is not easy even for CSC experts to determine CSC pluripotency of cells using bright field cell images alone, the deep learning of CSC enabled the AI viable to segment these cells. Furthermore, we examined the characteristics of the CSC-recognizing AI for a better deep-learning workflow.

## 2. Materials and Methods

### 2.1. Cell Culture

Cells of miPS-LLCcm derived from iPS-MEF-Ng-20D-17 provided by the Riken BioResource Research Center through the National BioResource Project of the Ministry of Education, Culture, Sports, Science, and Technology/Japan Agency for Medical Research and Development, Japan [12], were cultured in a medium containing Lewis lung cancer (LLC) cell culture-conditioned medium (cm) mixed with Dulbecco’s Modified Eagle Medium (DMEM) high glucose with 15% fetal bovine serum (FBS), 1× non-essential amino acids (NEA), 1% penicillin/streptomycin (P/S), 1× L-glutamine, and 100 μM of 2-mercaptoethanol in a ratio of 1:1 on 96-well plates (Sumitomo Bakelite, Tokyo, Japan) precoated with 0.1% porcine-skin gelatin (MilliporeSigma, St. Louis, MO, USA) following the procedure described in a previous report [3]. We grew LLC cells in DMEM high glucose supplemented with 10% FBS, 1× NEA, and 1% P/S. To obtain the cm, we maintained confluent grown cells in the medium by changing 10% FBS to 5% for 1 day; subsequently, we filtrated the cm. We sustained the cells at 37 °C in a 5% CO_2_ incubator (PHC, Gunma, Japan). The culture reagents were purchased from FUJIFILM Wako Pure Chemical Corporation, Osaka, Japan.

### 2.2. Microscopy

We observed cells grown in 96-well plates under a fluorescence microscope BZ-X800 (KEYENCE, Osaka, Japan) equipped with a CFI Plan Fluor DL 10× lens (Nikon, Tokyo, Japan). We acquired a GFP fluorescence image combined with a corresponding phase contrast image for each well. All images had a resolution of 1920 × 1440 pixels in each channel. Images were saved in tiff format.

### 2.3. Image Processing and AI

We performed machine learning using a personal computer equipped with one CPU, 32 GB of RAM, and one GPU (NVIDIA Corp., Santa Clara, CA, USA). We performed CGAN using pix2pix [13] ported to TensorFlow deep learning environment (https://github.com/affinelayer/pix2pix-tensorflow), obtained on 17 May 2019, built in the Linux OS. We processed a pair of images obtained from microscopy having a resolution of 1920 × 1440 pixels to make 35 new image pair files, with each image having a resolution of 256 × 512 pixels. The paired images were used as input to the pix2pix. We calculated the F-measure value to evaluate the AI output against the target. The color images were transformed to black and white images; the images in the pair were overlaid to find an identical region; and the identical region was employed for the calculation of precision and recall. We defined the F-measure using Equation (1), as: (1)$$F-\mathrm{measure}=\frac{2\mathrm{RP}}{R+P}$$
where *R* is the recall and *P* is the precision.

We performed CNN classification in Chainer and PyTorch environments built in the Linux OS. Gradient-weighted class activation mapping (Grad-CAM), guided Grad-CAM, and Grad-CAM++ were used to obtain visual explanation of CNN classification [14,15].

### 2.4. Statistical Analysis

Single-factor analysis of variance (ANOVA) was used for the evaluation of group distribution. Furthermore, Scheffe’s F test was performed to evaluate the differences between the two groups.

## 3. Results

### 3.1. Image Acquisition of Cultured CSC and Deep Learning

As a CSC model, we used miPS-LLCcm cells [3]. The cell retains diverse SC marker expression as a normal miPS cell with a novel potency difference of forming a malignant tumor when transplanted to a mouse. The characteristics were extensively studied. The cell harbors green fluorescence protein (GFP) reporter gene under Nanog promoter, which is one of the essential SC marker genes [12]. Monitoring SC pluripotency of the cell can be simply performed based on the presence of GFP fluorescence. Only cells with GFP fluorescence are determined to be CSC with confidence. Unsurprisingly, some cultured cells lost the fluorescence during the cultivation, owing to the capability of the CSCs to differentiate. Its purpose was to indicate that AI could learn the location of CSCs through cell morphology. We used a fluorescence image of GFP as the supervisor for deep learning using CGAN. We expected the AI to show CSC in phase-contrast cell images without GFP fluorescence information. The construction of the four types of AI was accomplished using different image datasets to evaluate the limitation of AI regarding different cultivation periods (Figure 1a). However, no difference was observed among the four types of AI in the maintenance of expression of the Nanog-GFP reporter gene (Figure 1a). We observed diverse intensities of GFP fluorescence for each cell. In some cells, the fluorescence was nearly absent. The fluorescence property of the miPS-LLCcm cell was consistent with previous reports [3]. The result of dividing a large original image into smaller images was that some images contained no cells. We prepared datasets with and without such blank images. In deep learning using CGAN, the generator depicted a GFP fluorescence image from a phase-contrast image to trick the discriminator (Figure 1b). During deep learning, we trained the discriminator to find a fake image; however, it eventually became difficult to distinguish whether an image depiction was related to GFP fluorescence or paired GFP fluorescence to the phase-contrast image.

We used 1000 images for each AI training. The AI depicted a CSC image as an “output” from the phase-contrast image; the “input” from the remainder of the datasets comprising the selected images for the 1-day and 2-day cultures are shown as examples in Figure 2a,b, respectively. We observed diverse intensity of GFP fluorescence as shown by “target.” Some cells did not show the fluorescence, suggesting the loss of CSC characteristics. It was difficult to determine the difference between cells showing and losing GFP fluorescence in the phase-contrast images. The AI depicted the CSC images like the target (Figure 2a,b). There were examples where the contrast of the AI-depicted CSC image resembled those by GFP fluorescence intensity shown in the target (Figure 2b). CSC images provided by AI trained with selected images were not identical to those provided by AI trained without selection. We calculated recall, precision, and F-measure values to evaluate the similarity between the output and target (Figure 2c–e). The average values for recall and precision were at quite similar levels for all datasets, suggesting the AI models were built on valance between the values. By selecting images for training, the F-measure values significantly increased (Figure 2e). The maximum values ranged from 0.71 to 0.94 and the mean values ranged from 0.26 to 0.29. By inspecting each value and the corresponding images, high level values were observed a lot in pair images of the target and output, where a small number of cells showed GFP fluorescence.

### 3.2. Versatility of Application of AI to Datasets Other Than the Training Dataset

To examine the versatility of AI, we evaluated AI models using different kinds of test datasets that were not used to train the AI. Examples are shown in Figure 3a. The AI model of 1-day culture showed a mean F-measure value of 0.26 for the test dataset of the 1-day culture used as control (Figure 3b). Feasibly, we observed an approximately equal value for the test dataset of the 1-day-selected culture because the test dataset was essentially the same as the training dataset for the AI model. Interestingly, the value significantly decreased for both test datasets for the 2-day culture. The AI model of the 1-day-selected culture similarly showed a significantly low F-measure value of 0.26, while the control showed a value of 0.29 (Figure 3c). AI models trained with 1-day culture datasets might be incapable of depicting CSC images from the phase-contrast image of 2-day culture. In contrast, AI models of 2-day and 2-day-selected cultures showed similar F-measure values as the test datasets of 1-day and 1-day-selected cultures (Figure 3d,e).

It could be beneficial for AI to have the capability of depicting CSC images from phase-contrast images taken during any cultivation period. We trained the AI using a dataset comprising a mixture of 1-day and 2-day culture images for deep learning. The AI model trained with the mixed dataset of 1000 images showed a mean F-measure value of 0.29 for a test dataset containing a mix of 1- and 2-day culture images as control (Figure 4a). The values were almost the same for the other test datasets. Similarly, the AI model trained with mixed and selected datasets showed approximately the same values for all test datasets (Figure 4b). These results indicate the direction for improving the versatility AI by using images from a pool of several cultivation periods for deep learning.

### 3.3. CSC Object Recognized by Deep Learning

Because AI models of 1-day and 1-day-selected culture images showed lower performance on depicting CSC images than 2-day culture images, we hypothesized that these AI might recognize the difference in datasets between 1- and 2-day culture images. Thus, we examined whether we could classify cell images from 1-day and 2-day culture by deep learning using CNN (Figure 5a). For training AI, 3260 images with a resolution of 256 × 256 pixels from both 1-day and 2-day culture image datasets were subjected to data augmentation to increase datasets by changing the grayscale channel from one to three, randomly rotating plus/minus 30°, randomly excising the images by 224 × 224 pixels, and randomly flipping the images. To test the AI model, we subjected 100 images with a resolution of 256 × 256 pixels from both 1-day and 2-day culture image datasets to perform data augmentation by using a grayscale channel from one to three and randomly excising images by 224 × 224 pixels. We performed classification using transfer learning with the ResNet50 model. We observed that the loss value decreased immediately after two epochs and reached a value of approximately 0.22 for both training and test datasets at the end of training (Figure 5b, left). Inverse proportionally, the classification accuracy value increased and reached 0.91 (Figure 5b, right). We also observed good values through classification using transfer learning with the VGG16 model (Figure 5c). Both deep learning with ResNet50 and VGG16 succeeded in classifying cell images with good quality (Table 1).

Subsequently, we explored the referred region in the image to discriminate between the datasets of 1-day and 2-day culture. We applied Grad-CAM and improved Grad-CAM++ (Figure 5d) techniques to visualize the region. It highlighted the non-cellular region rather than the cellular region for the 1-day culture images using Grad-CAM. It also highlighted regions for 2-day culture images covered almost entirely with cells. In contrast to Grad-CAM, the highlighted regions were only cellular regions for the 1-day culture images using Grad-CAM++. Furthermore, CNN might be able to recognize cell density. Thereafter, we tested the CNN classification using images in which the masked image area by random erasing was 10–36%. However, we did not observe a marked decrease in the classification (data not shown), indicating that the CNN classification could utilize a value that had no cell density for the CSC recognition. Thus, it was difficult to obtain explicit knowledge of classification from the analysis.

We further examined other datasets consisting of only 1-day culture images in which each dataset had a difference, to provide better explanation of CSC recognition by AI models. Dataset of the 1-day-center consisted only of images prepared from the center of large original images whereas whole images were used in the dataset of 1-day culture. This was previously shown to have a significant increase in the F-measure values against the dataset for the AI model of CGAN [5]. Thereafter, we performed CNN classification using 400 images that were split into 10 and 390 images for test and training, respectively. After data augmentation, we performed classification using transfer learning with the VGG16 model. We observed that the accuracy value increased and it reached 0.8 after 20 epochs, suggesting good classification between the center-selected images and whole images. Grad-CAM highlighted cells instead of non-cellular space during the classification of center-selected images (Figure 5e). Interestingly, guided Grad-CAM, which is a tuned Grad-CAM with high-resolution, highlighted approximately the edge of the cells for the 1-day culture without selection while the whole cell with intra-cellular structures was highlighted for the center images. Because both datasets consisted of the same 1-day culture images, the observed structure by guided Grad-CAM might be owed to the optical effect of the microscope.

## 4. Discussion

We investigated the property of AI models trained to recognize the morphology of cultured CSC expressing the Nanog-GFP reporter gene using CGAN. The AI models had the capability to depict CSC images with high accuracy through F-measure values. However, average F-measure values were relatively low. To improve the efficiency, we found several points that needed reminding for training the AI models. Blank images in the AI training dataset reduced the F-measure value. The F-measure values tended to reduce when an AI model was tested for cell images with a longer cultivation period than when using the images employed for training the AI model. By contrast, an AI model trained using cell images of a long cultivation period did not show such inefficiency to cell images of short cultivation. Therefore, an improved workflow for training AI consists of using a pool of cell images ranging from short to long cultivation periods, regardless of including blank images, for improving efficiency.

The application of deep learning is emerging on cell morphology and structures [16]. The performance of the described AI models was extraordinary. A well-annotated dataset of cell images can accelerate AI technologies in microscopy cell analysis [17]. In contrast to those cells used for the study, attempts to distinguish CSC from non-CSC using AI are scarce because experts have been unable describe the difference clearly. CSC-like cells from human nasopharyngeal carcinoma showed squamous morphology, same as the non-CSCs [18]. Both cell types changed their morphology after long culture. The other example is in miPS-LLCcm cells. Cells transform from CSCs to non-CSCs with apparent change of morphology [3]. According to the previous study in which CSC was predicted from cell morphology, this study showed the further possibility that information on the extracted images during deep learning might help the workflow of constructing AI using CGAN. Although the visualization technology of CNN classification showed regions where AI is recognized, the information required more analysis to improve the workflow.

The nuclear architecture of cells is recognized to undergo morphological changes when SC differentiates [19]. For example, the nuclear position of the human chromosome encoding Nanog moved to central when compared with differentiated cells [20]. By ordering the chromatin structure, the transcriptional molecular machinery of SC-specific genes is exposed or sheltered. Thus, we often label nuclear bodies with fluorescent dyes for the observation. Interestingly, the human iPS cell colony was discriminated from non-iPS cell colony using a supervised machine-learning algorithm [21]. The classification of two cell types largely depends on the inside edge of the colony. Demonstrating a difference in nuclear localization for nuclear bodies between the cell types suggested that the AI recognition was marked using information from the nuclear substructure. By differentiation, miPS cell also showed characteristic morphology similar to hiPSC [2]. The feature included a dropping round shape, large nucleoli, scant cytoplasm, and unclear edge of the cell. We observed changes of the image from undifferentiated to differentiated by shining on the dark under phase-contrast microscopy. By contrast, miPS-LLCcm cells were composed of several kinds of CSC transformed from SCs with various ranges changing the normal features. The morphological features of miPS-LLCcm are not the same as in the case of miPS cell. However, the AI recognition of cells shown by Grad-CAM analysis was consistent with the previous report on hiPSC. Our CNN classification of CSC might enable the extraction of informational signals of structure that were difficult to distinguish even for an SC expert under a microscope.

AI is expected in cancer diagnosis and the related studies. This study implied the importance of training datasets being used from several days of cultivation. Our approach to achieve an AI model could also be useful for CSCs other than miPS-LLCcm. Although we showed the usefulness of CGAN for deep learning of CSC morphology, the values evaluating AI models were low. To improve AI models, further studies are required in the analysis, using available technology by utilizing datasets consisting of only CSC images and non-CSC images. Because miPS-LLCcm cells grow crowded and often lie on top of other cells, it is difficult to identify their cell boundaries. It is interesting to train AI to not use datasets of a mixture of CSCs and non-CSCs, but rather, separated datasets at the individual cell level.

## 5. Conclusions

We evaluated the versatility of AI models using CGAN trained to map undefined live CSC morphology. When the AI model was applied to live CSCs, it was efficient in using CSC images from various culture conditions. The visualization technology of deep-learning progress of CNN classification of CSC was powerful to obtain insights into how outputs of AI models are produced. However, it showed that objects used to differentiate classes were not always cells but also the instruments and microscopic effect of the view. The improved deep-learning workflow indicated in this study could be useful for the development of an AI model identifying undescribed morphological characteristics in CSCs.

## Figures and Tables

**Figure 1 biomedicines-10-00941-f001:**
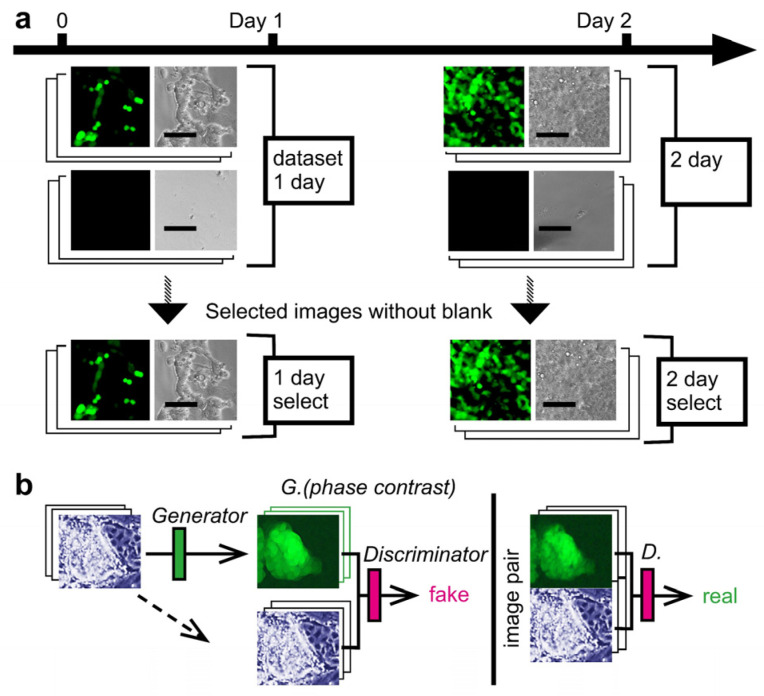
Experimental design. (**a**) Cells image datasets for deep learning. Cell images obtained were from cultured plates on days 1 and 2 to prepare 1000 pairs per dataset. Datasets were used with or without blanks for deep learning. Datasets of 1-day and 2-day included blank image pairs of 158 and 102, respectively. Bars = 100 μm. (**b**) The CGAN was trained to map grayscale bright-field cell images into color dark-field fluorescence images. Learning was performed for 200 epochs, with a batch size of one per epoch.

**Figure 2 biomedicines-10-00941-f002:**
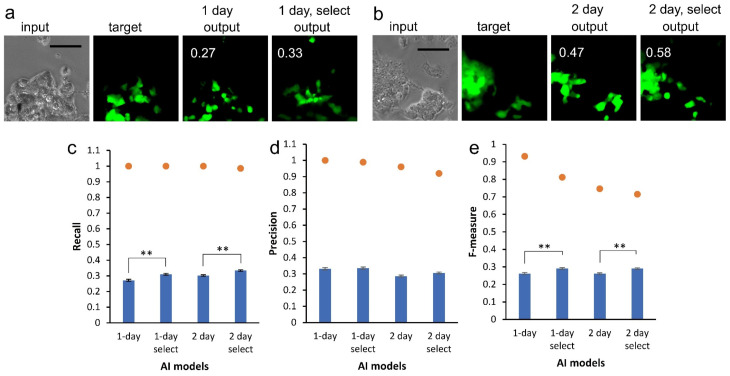
CSC image mapping from bright-field cell images. (**a**) Output examples by AI models trained with datasets of 1-day and 1-day-selected for an input phase-contrast image. Numbers in output images were F-measure values between output and target images of GFP fluorescence. Bar = 100 μm. (**b**) Output examples for datasets of 2-day and 2-day-selected. Bar = 100 μm. (**c**–**e**) Comparison of AI models trained with various datasets. Bright-field cell images chosen for AI to make outputs were taken from the same dataset category used for training the AI model. The values for recall (**c**), precision (**d**), and F-measure (**e**) were calculated. Closed circles indicate maximum values. Mean ± S.E.M., *n* = 1000. ** *p* < 0.01.

**Figure 3 biomedicines-10-00941-f003:**
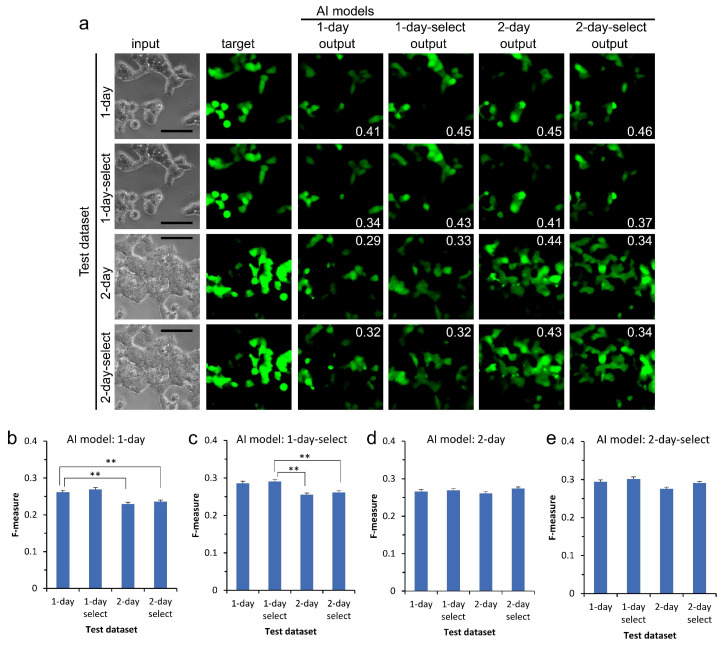
Indication of the versatility of AI models. (**a**) The AI models output examples for test datasets. Numbers in output images were F-measure values between output and target images. Bars = 100 μm. (**b**) Evaluation of the capability of AI was performed using F-measure values for test datasets (**b**–**e**). AI model trained using dataset of (**b**) 1-day, (**c**) on1e-day-selected, (**d**) 2-day, and (**e**) 2-day-selected. Mean ± S.E.M., *n* = 1000. ** *p* < 0.01.

**Figure 4 biomedicines-10-00941-f004:**
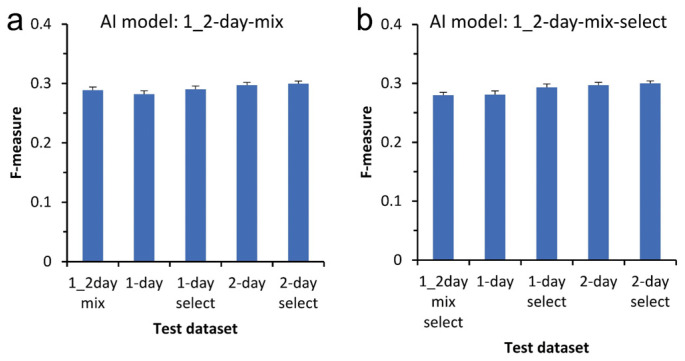
Effect of combining datasets for training AI. (**a**) AI model trained using a mixture of 1- and 2-day culture images datasets including 1000 image pairs. (**b**) AI model trained using the mixture of 1- and 2-day culture images dataset including 1000 pairs without blank images. Mean ± S.E.M., *n* = 1000.

**Figure 5 biomedicines-10-00941-f005:**
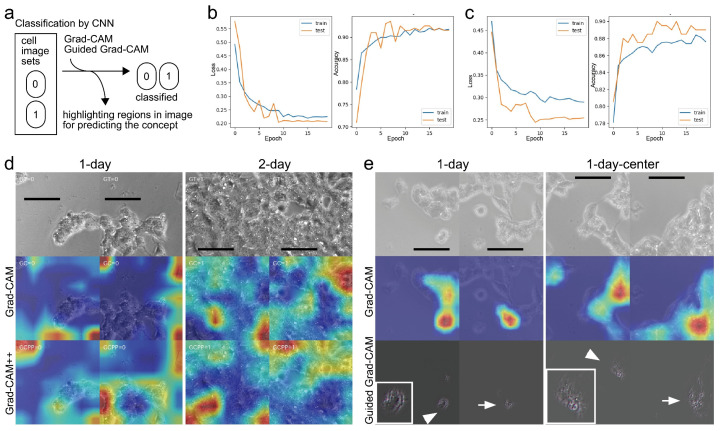
Classification of cells using CNN. (**a**) Scheme of CNN used to classify cell image datasets that were differently recognized by the AI model of CGAN. (**b**,**c**) Performing classification using transfer learning with (**b**) ResNet50 and (**c**) VGG16. Learning was performed in 20 epochs with a batch size of 100 per epoch for training datasets of 1- and 2-day cultures. (**d**) Visualization of CNN prediction. Output examples of cell images classified into 0 for a 1-day image and 1 for a 2-day image. Heat map from blue to red shows the magnitude of the probability of prediction from low to high using Grad-CAM and Grad-CAM++. Bars = 100 μm. (**e**) Visualization of CNN image classification into center and whole of a 1-day dataset. Grad-CAM prediction illustration was performed using the heat map. Arrow and arrowhead indicate prediction using guided Grad-CAM. Inset is a zoomed region shown by an arrowhead. Bars = 100 μm.

**Table 1 biomedicines-10-00941-t001:** Classification of CSC images of different culture periods using CNN.

Transfer-Learning Model	Class *	Precision	Recall	F-Measure
ResNet50	0	0.888	0.950	0.918
	1	0.946	0.880	0.912
VGG16	0	0.862	0.940	0.900
	1	0.934	0.850	0.890

* Class 0: 1-day dataset. Class 1: 2-day dataset.

## Data Availability

Image datasets used in this study are available from our Google drive (https://drive.google.com/drive/folders/1LypnjJ0DMi_42iMukHSZP5ncqjCGSmbJ?usp=sharing) on 18 April 2022.

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
