# Peer review of "Temporal and Locational Values of Images Affecting the Deep Learning of Cancer Stem Cell Morphology"

_biomedicines, 2022, doi:10.3390/biomedicines10050941_

Round 1
Reviewer 1 Report
The primary goal of this article was to evaluate the versality of AI models using CGAN trained to depict/map the CSC morphology and to differentiate them from non-CSC and to clarify its limitations. For cancer diagnosis and treatment, it is important to create a method to distinguish properly the CSC from the biopsies/smears and to achieve meaningful classification of cancer invasiveness based on the cell morphology/phenotyping.
The AI training was accomplished by using different datasets to evaluate the limitations related to the time-points of cell culture. Authors of these studies demonstrated that AI models trained on one-day CSC culture, which harbors GFP for learning purpose, could predict the microscopy images from two-day culture, but the quality factor was reduced. As stated by authors training datasets from one-day of culture are rather incapable to distinguish CSC from later days. Therefore, to achieve the correct classification the AI requires training datasets from several days.
Application of AI/deep learning is emerging in health-related studies and diagnosis. It is important to develop a proper approach to achieve high performance of AI technologies.
Publication of this article will be beneficial for the field. It presents the importance of right approach to achieve the correct diagnosis and future treatment patients.
Specific comments:
- It would be helpful to provide a list of abbreviations.
- Figure 2 – legend would be helpful.
Reviewer 2 Report
This manuscript is about the use of AI for recognizing cancer stem cells from a dataset of cells cultures.
The study seems interesting.
The manuscript is well written, however, the extensive use of acronyms makes it hard to read.
I have just some notes about the manuscript:
- line 31: "The findings of this paper can be applied to other cell types also. " The authors cannot claim this without further studies.
- 2.3. Image processing and AI: why do you use only the f-measure? why not present also the results for recall and precision? It would be easier to identify what needs to be improved.
- The results are really low, do you have any justification for this? And could they be improved?
- line 167: the maximum values are very different from the mean, do you think this is an outlier?
- About the datasets used, are they public and available?
- In general, the author should clarify the contributions of this work for its field of study.
